# High Sensitive Immunoelectrochemical Measurement of Lung Cancer Tumor Marker ProGRP Based on TiO_2_-Au Nanocomposite

**DOI:** 10.3390/molecules24040656

**Published:** 2019-02-13

**Authors:** Zheng Wei, Xiaoping Cai, Junping Zhang, Junming Fan, Jiangyan Xu, Liran Xu

**Affiliations:** 1Department of Oncology, Henan Academy Institute of Traditional Chinese Medicine, Zhengzhou 450000, Henan, China; questwz@163.com (Z.W.); caixiaoping987@126.com (X.C.); zhangjunping888@163.com (J.Z.); junmingfan@yeah.net (J.F.); 2School of Basic Medicine Sciences, Henan University of Chinese Medicine, Zhengzhou 450002, Henan, China

**Keywords:** progastrin-releasing peptide, immunosensor, small cell lung cancer, clinic diagnosis, titanium (IV) oxide microparticles

## Abstract

Progastrin-releasing peptide (ProGRP), which is known to be highly specific and sensitive to small cell lung cancer (SCLC), has been proven to be a valuable substitute for neuron-specific enolase in SCLC diagnostics and monitoring, especially in its early stages. The detection of ProGRP levels also facilitates a selection of therapeutic treatments. For the fabrication of our proposed biosensor, titanium (IV) oxide microparticles were first used, followed by dispersing gold nanoparticles into chitosan and immobilizing them onto a carbon paste electrode (CPE) surface. The developed immunosensor exhibits a much higher biosensing performance in comparison with current methods, when it comes to the detection of ProGRP. Therefore, the proposed CPE/TiO_2_/(CS+AuNPs)/anti-ProGRP/BSA/ProGRP is excellent for the development of a compact diagnostics apparatus.

## 1. Introduction

As a substance that can be found in blood, urine, or body tissues, tumor markers can be found in elevated levels in cancer, among other tissue types. Tumor markers are classified into different groups that indicate different disease processes. Therefore, these tumor markers have been applied in the detection of cancer occurrence in oncology. Elevated tumor marker levels are known to indicate that a patient has a tumor. Therefore, the determination of tumor marker levels is of vital significance for disease screening, diagnostics and prognostics [1,2]. Several main tumor markers have been extensively analyzed for the diagnosis of hepatocellular carcinoma, epithelial ovarian tumors, pancreatic cancer, colorectal cancer, and others, including human chorionic gonadotropin (hCG), prostate specific antigen (PSA), alphafetoprotein (AFP), carcinoma antigen 125 (CA125), carbohydrate antigen (CA19-9, CA15-3), and carcinoembryonic antigen (CEA). Around the world, studies have been carried out for the development and the improvement of clinical bioassays via affordable and portable diagnostic apparatuses [3,4,5,6]. It is essential for the protein biomarkers to be detected quantitively and sensitively, since their detection is significant across many fields, including biomedical research and diagnostics [7], systems biology [8] and proteomics [9]. Traditional methods of protein detection are enzyme-linked immunosorbent assays (ELISA) [10], radioimmunoassay (RIA) [11], electrophoretic immunoassay [12] and mass spectrometry-based proteomics [13]. However, they are lacking in sensitivity, time-consuming, expensive, and require complex instrumentation and substantial specimen volumes. Therefore, it is essential to develop low-cost, sensitive, rapid and facile techniques to detect protein for point-of-care treatments. Among the proposed biomarkers for small cell lung cancer (SCLC), neuron-specific enolase (NSE) is outstanding [14]. Compared with NSE, progastrin-releasing peptide (ProGRP) is more sensitive and specific in the detection of small cell lung cancer; thus, it has been used to complement NSE in the diagnosis and monitoring of SCLC, especially in its early stages [15]. Consisting of three isoforms expressed on the mRNA level, ProGRP is a precursor for neuropeptide gastrin-releasing peptide (GRP), generated in SCLC cells. Because only the total amount of ProGRP in SCLC has been measured, the potential functions and interchanging protein expressions remain to be studied.

Point-of-care treatments require sensitive, operationally facile, low limit of detection (LOD), and low-cost sensors that could provide rapid 3 multiplexed protein detection in the serum of both healthy individuals and cancer patients. The use of liquid chromatography–mass spectroscopy (LC-MS) for the detection of protein biomarkers has been reported on; some simpler techniques such as surface plasmon resonance [16,17], carbon nanotube based immunosensors [18], microcantilevers [19], nanowire transistor arrays [20] and nanocrystals [21], have also been proposed. However, these methods need to be multiplexed. Immunoassay, a common strategy for the quantitative detection of tumor markers, especially in cases of clinical laboratory use [22], remains significantly applicable to fields including biochemical [23,24,25,26,27] and environmental detection [28]. Unfortunately, the traditional methods suffer from having some inner drawbacks, including a lack of precision, requiring a prolonged period of time, and having difficulty in realizing automation. Therefore, it is essential to develop portable, semi-automated and compact immunosensors for rapid screening.

Considering the small size of nanocomposite materials compared with bulk materials, the former shows some different features, such as the quantum size effect, electrochemical features, catalytical features, and optical features. Thereby, they have gained acceptance for use in a wide range of applications, spanning many different fields, such as electrochemical biosensor fabrication. Gold nanoparticles (AuNPs) a form of nanomaterial that are ranked as the most applied metallic nanoparticles in electroanalytical uses. AuNPs have the potential to create a proper microenvironment, almost the same as that of redox protein, but with more freedom in orientation. In addition, the aforementioned nanoparticle has been reported to provide a direct charge transfer via the conductive channels of gold nanocrystals and then to decrease the insulation effect of protein shells. Moreover, the insulation shell of enzymes could be penetrated by the nanomeric edges of gold particles, which decreases the distance between the biomolecular redox sites and the electrode for charge transfer. It has been well acknowledged that titanium (IV) oxide (TiO_2_) nanocomposites are commonly used biocompatible materials due to their distinctive features, and have gained extensive use in the bioengineering and biomedical fields since they are nontoxic, chemically inert and strongly oxidizing [29,30]. They have also been reported to be used in the immobilization of biomolecules as a promising interface due to their eco-friendliness [31] and have gained extensive application in photochemistry [32] and electrochemistry [33,34,35]. A carbon paste electrode (CPE) is made from a mixture of conducting graphite powder and a pasting liquid, which can offer an easily renewable surface for electron exchange and is widely used, mainly for voltammetric measurements.

The present work features a TiO_2_, AuNPs via chitosan (CS) -modified CPE, whereby a sensitive and renewable immunosensor (CPE/TiO_2_/(CS+Au)) was successfully prepared to determine the model reagent ProGRP. The analysis of ProGRP levels in adults is of vital importance for the diagnosis of lung cancer at early stages. Therefore, the ProGRP immunosensor configuration has the potential to provide convincing results. For a biochemical analysis, the major analytical techniques include the immunoassay methods based on the highly specific molecular recognition of antigens by antibodies [36,37]. In our case, the formation of carcinoembryonic antigen (CEA) antibody–antigen complexes on the final electrode was probed via the [Fe(CN)_6_]^3−/4−^ redox pair and monitored by cyclic voltammetry (CV) and electrochemical impedance spectroscopy (EIS) measurements. Therefore, the developed immunosensor could be used for the determination of ProGRP in synthetic serum specimens.

## 2. Results and Discussion

The electrochemical properties of each assembly step of the modified electrode surface were investigated using CV and EIS measurements. Figure 1 showed the CV characterizations of the modified CPEs in a 1 mM [Fe(CN)_6_]^3−/4−^ solution, whereas Figure 2 displays the EIS curves. As shown in Figure 1, the plain CPE/TiO_2_/(CS+AuNPs) exhibited the maximal peak current. An obvious decrease in the peak current was observed after the anti-ProGRP was immobilized onto the surface of the electrode (CPE/TiO_2_/(CS+AuNPs)/anti-ProGRP), which indicated that the active sites and effective area for the charge transfer were decreased. The decrease in active sites and effective area was due to the surface coverage of anti-ProGRP, which hid the exposure of TiO_2_ and AuNPs. In addition, the nonconducting state of the anti-ProGRP also decreased in response to [Fe(CN)_6_]^3−/4−^, during the CV scan. Albumin from bovine serum (BSA) was used to block the remaining active sites of the electrode surface. The interaction between the antibody and the antigen was completed after immersing the proposed immunosensor into the ProGRP solution (CPE/TiO_2_/(CS+AuNPs)/anti-ProGRP). Considering the immunocomplex reaction of this configuration, the peak current was dramatically decreased. The schematic diagram of the sensor preparation is illustrated in Scheme 1.

The EIS spectra recorded for the different electrodes consisted of semicircle segments and a linear segment. The electron-transfer resistance (*R*_et_), referred to the diameter of the former segment at higher frequencies whereas the diffusion process denoted the latter segment at lower frequencies. The EIS experiment was used to investigate the stepwise immunosensor fabrication. The calculation of the semicircles was based on the Randle’s cirquit. The phase angle of linear segments fitted well with the Warburg model. The EIS measurements of different modified CPE at various stages were shown in Figure 2, and the inset showed the optimized fitting circuit model [38]. An insignificant semicircle at high frequencies and a linear segment at low frequencies were observed in the Nyquistic diagram for the CPE/TiO_2_/(CS+AuNPs), which indicated an extremely decreased *R*_et_ to redox probe [Fe(CN)_6_]^3−/4−^. An apparent increase in resistance for the redox probe was observed after the anti-ProGRP was immobilized, which demonstrated the distinctive property of the anti-ProGRP—electric insulation (the formation of a barrier for the charge transfer on the surface of the electrode), along with the completion of the modification process of CPE/TiO_2_/(CS+AuNPs)/anti-ProGRP. This was followed by immersing the BSA-blocked CPE/TiO_2_/(CS+AuNPs)/anti-ProGRP/BSA into the ProGRP solution. The maximal *R*_et_ was observed at CPE/TiO_2_/(CS+AuNPs)/anti-ProGRP/BSA/ProGRP, suggesting a desirable antibody-antigen reaction.

In spite of the convincing results provided by CVs and EIS spectra for the electrode modification, the CV measurement offered a more obvious drop than the EIS experiment. The results showed that CV exhibited a better performance in the characterization of modification in comparison with EIS, for the proposed CPE.

As shown in Figure 3, the impedimetric response of our proposed electrode was recorded as a function of the ProGRP concentration over a range of 10 to 500 ng/mL in PBS, that contained [Fe(CN)_6_]^3−/4−^ for an incubation period of approximately 5 min. As the concentration of ProGRP was increased, a capacitance decrease was observed, possibly due to the varied dielectric/blocking features of the electrode–electrolyte interface, caused by the interaction between the antigen and the antibody. Considering the interaction between ProGRP and the anti-ProGRP, the capacitance decrease was expected to result from the distance increase between the electrolyte and the electrode. Furthermore, the capacitance magnitude decreased due to the interaction between ProGRP and the anti-ProGRP, which in turn was caused by the decrease in polar ProGRP protein molecules that replaced the water molecules on the surface of the electrode. The variation of capacitance with the concentration of analyte was plotted for the calibration of the capacitive immunosensor, with the linear relationship, obtained as 10–500 ng/mL. The variation in the capacitance of our proposed electrode was used as a function of the ProGRP concentration under similar conditions, with a linear regression coefficient (*r*^2^) of 0.991, shown for this sensor. The developed sensor was highly sensitive, possibly due to the increased surface area and the functional features of the CPE/TiO_2_/(CS+AuNPs). The biosensor provided an excellent and effective performance, due to the increased loading capacity for antibody molecules, caused by the available functional groups present on the TiO_2_/(CS+AuNPs). Furthermore, the developed sensor showed a low LOD of 0.133 ng/mL (3*σ*_b_/*m*).

The specificity of CPE/TiO_2_/(CS+AuNPs)/anti-ProGRP/BSA/ProGRP was investigated by incubating this immunoelectrode with 15 μL of pathogens and metabolites, including 300 ng/mL of ProGRP, 5 mM of ascorbic acid, 5 mM of uric acid, and 5 mM of glucose. As shown in Figure 4, after these pathogens were added into the proposed electrode, the electrochemical response showed no variations during the experiments, which suggested that the probe for ProGRP was highly specific.

For the investigation of the reproducibility of the immunoelectrode, ProGRP (300 ng/mL) was used with the proposed electrode, while the variation in the current magnitude was recorded. As shown in Figure 5A, no apparent variation was observed, which suggested that this electrode was highly precise. The CVs (Figure 5B) of our proposed immunoelectrode were recorded for ca. 30 d (10 d intervals) to study its storage stability. During the 30 days, no apparent current variation was observed; afterwards, the current showed a 4% variation from the original value. These results indicated a remarkable storage stability of our proposed electrode, for as long as 30 d.

For the investigation of the practical performance, three serum samples were analyzed in both the enzyme-linked immunosorbent assay (ELISA, Cusabio) method and the proposed electrochemical immunosensor. ELISA is a recommended technology, commercially used for the ProGRP test. An antibody specific to ProGRP was first coated on the microplate. Then, samples were added to the microplate with a biotin-conjugated antibody. The antigen was combined with the antibodies in the microplate by a competitive binding test. Next, avidin conjugated to horseradish peroxidase (HRP) and tetramethylbenzidine (TMB) were added, and formed a color change. The result was measured spectrophotometrically at a wavelength of 450 nm. For a comparison test, the standard addition method was used. The results are summarized in Table 1. As shown in the table, the ELISA could not detect ProGRP above 2 ng/mL, while the proposed immunosensor showed excellent performance at high-concentration conditions.

## 3. Experiments

### 3.1. Chemicals

ProGRP and anti-ProGRP were commercially available in Shengyuan Biotech Inc., Shenzhen, China. All test reagents were of analytical grade. For all experiments, the supporting electrolyte was a pH 7 KH_2_PO_4_ buffer solution (PBS), where 1 M NaOH was added to adjust the pH value. The following were purchased from XFNANO Co. Ltd (Nanjing, China): Acetic acid; commercial AuNPs; CS monomer, for the preparation of TiO_2_NPs, Ti(OCH(CH_3_)_2_)_4_; absolute ethanol, for synthetic serum preparation of NaCl; CaCl_2_; *N*-Ethyl-*N*-(3-dimethylaminopropyl) carbodiimide (EDC); *N*-hydroxysuccinimide (NHS); KCl; urea, which was commercially available in Sigma-Aldrich (Shanghai, China); and Au nanoparticles with an average size of 20 nm (0.05 mg/mL).

### 3.2. Nanocomposite Preparation

For the preparation of the TiO_2_/(CS+Au)-modified CPE, first TiO_2_ (20 %), mineral oil (30 %), and graphite powder (50 %) were mixed together to prepare TiO_2_/CPE. The yielded paste was filled into the hole of a Teflon body (radius: 2 mm), and the electrical contact for this CPE was a copper wire. The CS flakes were dissolved into an acetic acid solution (50 mL, 2 M) which was added with the commercial AuNPs (2 µL) and was stirred for 60 min. This was followed by a 15 min sonication, so as to uniformly disperse the AuNPs into the CS. Subsequently the CS/AuNPs solution (25 µL) was uniformly spread onto the TiO_2_/CPE. After drying in a desiccator for 45 min, the TiO_2_/(CS+AuNPs)/CPE was yielded.

### 3.3. Immunosensor Fabrication

The immunosensor was prepared by firstly immersing TiO_2_/(CS+AuNPs)/CPE into the anti-ProGRP solution for 60 min at 4 °C. The unspecific bindings were prevented through blocking the remaining active sides on the CPE surface with 0.25 % (*w*/*w*) BSA for 20 min. This was followed by washing the electrode surface using distilled water. The TiO_2_/(CS+AuNPs)/anti-ProGRP/BSA/CPE immunosensor was stored at 4 °C in pH 7 phosphate buffer solution (PBS), prior to use.

### 3.4. Measurement

The electrochemical experiment was carried out with an Autolab Potentiostat/Galvanostat (PGSTAT302/N, Metrohm), where a three-electrode configuration was used. The references and the auxiliary electrode were Ag/AgCl and a platinum wire, respectively. The EIS characterization was performed using a multi-impedance test system with a frequency range of 10 kHz–10 mHz and an AC amplitude of 10 mV.

## 4. Conclusions

In the present work, ProGRP was successfully detected through the interactions between antibodies and antigens using a CPE/TiO_2_/(CS+AuNPs) nanohybrid-based immunosensor. The electrochemical response for the CPE/TiO_2_/(CS+AuNPs)/anti-ProGRP/BSA/ProGRP immunoelectrode, as a function of the ProGRP concentration, indicated that this electrode was highly sensitive, displayed long-term stability, and possessed a low detection limit. Therefore, the proposed immunosensor could be potentially used in the medical diagnostic field.

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
