# Peer review of "High Sensitive Immunoelectrochemical Measurement of Lung Cancer Tumor Marker ProGRP Based on TiO2-Au Nanocomposite"

_molecules, 2019, doi:10.3390/molecules24040656_

Round 1

Reviewer 1 Report

The MS "molecules-420671" entitled "High sensitive immunoelectrochemical measurement of lung cancer tumor marker ProGRP based on TiO2-Au nanocomposite" is well written, topical, and of sufficient scientific importance. Al in all some important part regarding real use of designed biosensor is missing.  Therefore I recommend that MS could be published whether it meets the technical standards for publication in Molecules after major revision.

Comments:

1, In section 2.4. - There should be mentioned the type of Potentiostat/galvanostat from Autolab including producer ful name and country of origin. In addition not all the instruments from Autolab contains impedance module.

2, Pro GRP could be determined using standard ELISA method, what is real advantage of developed biosensor against it?

3, There is missing part where the designed biosensor is applied on a real sample. It should be also compared with ELISA when it would be done.

4, There is also missing at least small mention about what is the target sample, and what is recommended methodology or tested procedure for handling with it.

5, Any schematic drawing of the concept is needful to show how the biosensor complex works and where is the bottleneck of selectivity of presented biosensor.

6, Similar article dedicate to this topic was published in 2011 by Zhou et. Al. (DOI: 10.1016/j.bios.2011.02.043) and is not cited in the MS.

List of ambiguities in text:

row 19,….apparatuses (apparatus)

row 57,...suffer (insert "by") some inner...

row 77,  The presented work present… (replace “present“ by "shows")

row 83, ...ormation / ("formation")...

row 96, 2.2.(gap is missing)Nanocomposite preparation…

row 100, And CS flakes… (replace “And” by “The”)

row 179, …pathogensw ere… (replace by “pathogens were”)

row 205, …province (remove surplus gaps)in China…

Author Response

1, In section 2.4. - There should be mentioned the type of Potentiostat/galvanostat from Autolab including producer full name and country of origin. In addition, not all the instruments from Autolab contains impedance module.

Reply: Thank you for your comment. We added the Autolab model (PGSTAT302N) in the revised manuscript.

2, Pro GRP could be determined using standard ELISA method, what is real advantage of developed biosensor against it?

Reply: Electrochemical immunosensor is a kind of labeled immunoassay method which combines electrochemical detection technology with immune technology. Among them, the performance of the current immunosensor is particularly outstanding. Because many properties of nanomaterials can be used to improve the sensing signal, the sensitivity of electrochemical immunosensor has been greatly improved. Electrochemical immunosensing has the potential to be developed as a chip sensor. The sensor is made into a chip plus a small device, which can be operated via a phone or laptop. The electrical signal is converted into computer language to get the result, which is very suitable for point of care.

3, There is missing part where the designed biosensor is applied on a real sample. It should be also compared with ELISA when it would be done.

Reply: We acknowledge the suggestion. We further conducted a real sample test and compared with the ELISA method. The result has been presented in Table 1 in the revised manuscript.

4, There is also missing at least small mention about what is the target sample, and what is recommended methodology or tested procedure for handling with it.

Reply: We acknowledge the comment. We further added some description of commercial recommended ELISA procedure in the revised manuscript. 

5, Any schematic drawing of the concept is needful to show how the biosensor complex works and where is the bottleneck of selectivity of presented biosensor.

Reply: We acknowledge the comment. We further added a scheme in the revised manuscript.

6, Similar article dedicate to this topic was published in 2011 by Zhou et. Al. (DOI: 10.1016/j.bios.2011.02.043) and is not cited in the MS.

Reply: We acknowledge the comment. We further cited two relative articles (include the Zhou et al.) in the revised manuscript.

List of ambiguities in text:

row 19,….apparatuses (apparatus)

row 57,...suffer (insert "by") some inner...

row 77,  The presented work present… (replace “present“ by "shows")

row 83, ...ormation / ("formation")...

row 96, 2.2.(gap is missing)Nanocomposite preparation…

row 100, And CS flakes… (replace “And” by “The”)

row 179, …pathogensw ere… (replace by “pathogens were”)

row 205, …province (remove surplus gaps)in China…

Reply: Thank you for pointing out the language mistakes in our manuscript. We corrected these errors in the revised manuscript.

Reviewer 2 Report

This work is about of electrode preparation as a biosensor for detection of Progastrin-releasing peptide (ProGRP), that is known to be highly specific and sensitive to small cell lung cancer (SCLC). The modified electrode is base on carbon paste electrode (CPE) with titanium dioxide, gold nanoparticles dispersed into chitosan which is immobilized on CPE surface.

English language and style are minor spell check required

In the introduction, there is no talk of using carbon paste electrodes and why that choice. Could they also be modified electrodes such as vitreous carbon, ordinary pyrolytic graphite? For example

Why pH 7 and not the physiological one?

There is no mention of the size of the gold nanoparticles or the origin.

The preparation of the base electrode is clear. The preparation with the biosensor is not clear. This is explained in detail in figure 1.

The nomenclature of the electrodes is not clear. We talk about TiO2/(CS + AuNPs)/anti-ProGRP/BSA/CPE and after ProGRP/TiO2/(CS + AuNPs). Is it the same electrode? And then we talk about the immobilization of anti-ProGRP.

In figure 1 given CPE/TiO2/(CS + AuNPs)/anti-ProGRP/BSA/ProGRB, CPE/TiO2/(CS + AuNPs) and CPE/TiO2/(CS + AuNPs)/anti-ProGRB

Later in the text talks about TiO2/(CS + AuNPs)/anti-ProGRP/BSA/ProGRP/CPE

Where did the cyclic voltammetry start in Figure 1?

The explanation of the decrease in the current in the cyclic voltammetry of the [Fe(CN)6]3-/4- is not clear. It may be an effect of the change in the surface area and the active site, but the change in the area and the active site is not explained.

the cyclic voltammetry is the first measurement? Or are it stabilization?

In figure 4, the beginning of the sweep is not indicated. On the X-axis there is an error in the unit of measurement. It must be V and not mV.

Are reversible the processes shown in Figure 4? Organic acids are usually irreversible processes. These measurements are in the presence of [Fe(CN)6]3-/4-?

Figure 5 does not include error markers.

In figure 5 the current must be in mA in the Y-axis.

There is only EIS measurement to different concentrations. It is more used current versus concentration.

Author Response

English language and style are minor spell check required

Reply: Thank you for your comment. We carefully checked typos in the revised manuscript.

In the introduction, there is no talk of using carbon paste electrodes and why that choice. Could they also be modified electrodes such as vitreous carbon, ordinary pyrolytic graphite? For example

Reply: Thank you for your comment. We are sorry for making you confused. We added some description of CPE in the revised manuscript. A carbon paste electrode (CPE) is made from a mixture of conducting graphite powder and a pasting liquid, which can offer an easily renewable surface for electron exchange and widely used mainly for voltammetric measurements. Reviewer mentioned vitreous carbon such as glassy carbon electrode also can be used for construction of electrochemical immunosensor.

Why pH 7 and not the physiological one?

Reply: Thank you for your comment. We understand the physiological condition (pH=7.4) is more suitable for biological analysis. In this case, the signal of ferrocene has been used for representing the concentration of ProGRP. The electrochemical response of ferrocene at pH=7 is stable and well-studied. Therefore, pH 7 has been selected in this work.

There is no mention of the size of the gold nanoparticles or the origin.

Reply: Gold nanoparticles used in this work were purchased from XFNANO Co.Ltd. The average size of the gold nanoparticles is 20 nm. The concentration is 0.05 mg/mL. The detail information has been added in the revised manuscript.

The preparation of the base electrode is clear. The preparation with the biosensor is not clear. This is explained in detail in figure 1.

The nomenclature of the electrodes is not clear. We talk about TiO2/(CS + AuNPs)/anti-ProGRP/BSA/CPE and after ProGRP/TiO2/(CS + AuNPs). Is it the same electrode? And then we talk about the immobilization of anti-ProGRP.

In figure 1 given CPE/TiO2/(CS + AuNPs)/anti-ProGRP/BSA/ProGRB, CPE/TiO2/(CS + AuNPs) and CPE/TiO2/(CS + AuNPs)/anti-ProGRB

Later in the text talks about TiO2/(CS + AuNPs)/anti-ProGRP/BSA/ProGRP/CPE

Reply: We are sorry for making you confused. The nomenclatures of the electrodes were carefully corrected (both in content and figure caption). Additional schematic diagram was included in the revised manuscript. 

Where did the cyclic voltammetry start in Figure 1?

Reply: The cyclic voltammetry started at -0.4 V. We further added an arrow in the Figure 1.

The explanation of the decrease in the current in the cyclic voltammetry of the [Fe(CN)6]3-/4- is not clear. It may be an effect of the change in the surface area and the active site, but the change in the area and the active site is not explained.

Reply: We agree with the point raised by the reviewer. The decrease in active sites and the effective area is due to the surface coverage of ProGRP, anti- ProGRP and BSA, which hided the exposure of TiO2 and AuNPs. In addition, the non-conducting state of the ProGRP and its antibody also decreases in [Fe(CN)6]3-/4- response during the CV scan. We revised the manuscript accordingly.

The cyclic voltammetry is the first measurement? Or are it stabilization?

Reply: The cyclic voltammograms presented in Figure 1 are the second scan because the first scan could still remain some surface state change.

In figure 4, the beginning of the sweep is not indicated. On the X-axis there is an error in the unit of measurement. It must be V and not mV.

Reply: Thank you for pointing out the error. We revised Figure 4 accordingly.

Are reversible the processes shown in Figure 4? Organic acids are usually irreversible processes. These measurements are in the presence of [Fe(CN)6]3-/4-?

Reply: These measurements were performed in the presence of [Fe(CN)6]3-/4-. In fact, the signal of [Fe(CN)6]3-/4- was used for measuring the concentration of ProGRP. The direct redox peaks of the interference species are unobservable during the scan due to the high response of the [Fe(CN)6]3-/4-.

Figure 5 does not include error markers.

In figure 5 the current must be in mA in the Y-axis.

Reply: We acknowledge the suggestion. Figure 5 has been revised accordingly. 

There is only EIS measurement to different concentrations. It is more used current versus concentration.

Reply: We understand reviewer’s concern. The EIS measurement is quite complicated than that of the current measurement. However, we found the EIS measurements in this work are much reliable than that of the CV scans. Therefore, we chose EIS for ProGRP concentration determination.

Round 2

Reviewer 1 Report

All the comments were satisfactorily addressed.